# External validity of machine learning-based prognostic scores for cystic fibrosis: A retrospective study using the UK and Canadian registries

**Yuchao Qin** [1] *, **Ahmed Alaa**[2,3], **Andres Floto**[1], **Mihaela van der Schaar**[1,4,5]

**1** University of Cambridge, Cambridge, United Kingdom, **2** University of California Berkeley, Berkeley, California, United States of America, **3** University of California San Francisco, San Francisco, California, United States of America, **4** Alan Turing Institute, London, United Kingdom, **5** University of California Los Angeles, Los Angeles, California, United States of America

* yq257@cam.ac.uk

## Abstract

Precise and timely referral for lung transplantation is critical for the survival of cystic fibrosis patients with terminal illness. While machine learning (ML) models have been shown to achieve significant improvement in prognostic accuracy over current referral guidelines, the external validity of these models and their resulting referral policies has not been fully investigated. Here, we studied the external validity of machine learning-based prognostic models using annual follow-up data from the UK and Canadian Cystic Fibrosis Registries. Using a state-of-the-art automated ML framework, we derived a model for predicting poor clinical outcomes in patients enrolled in the UK registry, and conducted external validation of the derived model using the Canadian Cystic Fibrosis Registry. In particular, we studied the effect of (1) natural variations in patient characteristics across populations and (2) differences in clinical practice on the external validity of ML-based prognostic scores. Overall, decrease in prognostic accuracy on the external validation set (AUCROC: 0.88, 95% CI 0.88-0.88) was observed compared to the internal validation accuracy (AUCROC: 0.91, 95% CI 0.90-0.92). Based on our ML model, analysis on feature contributions and risk strata revealed that, while external validation of ML models exhibited high precision on average, both factors (1) and (2) can undermine the external validity of ML models in patient subgroups with moderate risk for poor outcomes. A significant boost in prognostic power (F1 score) from 0.33 (95% CI 0.31-0.35) to 0.45 (95% CI 0.45-0.45) was observed in external validation when variations in these subgroups were accounted in our model. Our study highlighted the significance of external validation of ML models for cystic fibrosis prognostication. The uncovered insights on key risk factors and patient subgroups can be used to guide the cross-population adaptation of ML-based models and inspire new research on applying transfer learning methods for fine-tuning ML models to cope with regional variations in clinical care.

**Data Availability Statement:** Third-party registry data was used in this study, and the authors cannot legally distribute the data due to participant

confidentiality. Access requests for these registry data should be directed to the UK Cystic Fibrosis Trust and Cystic Fibrosis Canada. Instructions on data access application for the UK Cystic Fibrosis data can be found in this webpage: https://www.cysticfibrosis.org.uk/the-work-we-do/uk-cf-registry/apply-for-data-from-the-uk-cf-registry. The Canadian Cystic Fibrosis data we used is from Cystic Fibrosis Canada. Instructions on data access can be found in this webpage: https://www.cysticfibrosis.ca/our-programs/cf-registry/requesting-canadian-cf-registry-data.

**Funding:** YQ receives scholarship for his PhD study from the UK Cystic Fibrosis Trust. AF is funded by the US Cystic Fibrosis Foundation and the UK Cystic Fibrosis Trust (Digital Health Research Grant No. DHRP016). The funders had no role in the study design, data processing, model development and analysis, decision to publish, or preparation of the manuscript.

**Competing interests:** The authors have declared that no competing interests exist.

## Author summary

Cystic fibrosis is a genetic disease that affects multiple organs of a patient. End-stage cystic fibrosis patients usually have advanced lung diseases, and lung transplantation is the last treatment for them. Due to the scarcity of lung resources, precise and timely selection of high-risk patients for lung transplant referral is of paramount importance. Machine learning models were proved to be of potential in improving prognostic accuracy over existing clinical guidelines. However, the development of trustworthy machine learning models relies on a large volume of data. As a rare disease, cystic fibrosis only affects a small sub-population around the world. Reuse of well-established machine learning model developed from a large population is desirable. While prognostic scores of machine learning models were demonstrated effective within the same population, their external validity when applied to a demographically different patient cohort has not yet been fully explored. In this paper, we evaluated the external validity of a machine learning model with registry data of cystic fibrosis patients from UK and Canada and identified several risk factors and patient subgroups affected by variations across the two countries. These insights can be used to guide the cross-population adaptation of machine learning models in practice.

## Introduction

Respiratory failure caused by advanced lung disease is the most common cause of death among cystic fibrosis (CF) patients [1, 2]. Lung transplantation (LTx) is the last resort treatment for end-stage CF patients [3]. However, due to the scarcity of donor organs, LTx is only assigned to patients most at risk with a priority scheme (waitlist) that is based on estimates of individual patient prognosis. Current guidelines of LTx referral typically rely on the forced expiratory volume in one second ($FEV_1$) metric, which is a measurement of lung function and a strong predictor of CF patients' mortality [4], as a surrogate for patient risk and a predictor of individual patient outcomes. In the new era of personalized medicine [5], it has been shown that more precise estimates of individual patient risk can be obtained using machine learning-based prognostic models that incorporate multiple predictors beyond the $FEV_1$ biomarker [6]. Such prognostic models can achieve higher accuracy in identifying appropriate patients for LTx referral as opposed to the $FEV_1$-based baseline.

Despite their potential for improving prognostic accuracy, training ML models typically requires a large number of data points—the scarcity of data in rare diseases such as CF could compromise the accuracy and validity of such models. Since CF only affects a small sub-population worldwide, with varying incidence rates across different regions [7], the lack of data may hinder the development of robust and reliable ML models in areas with smaller populations or lower CF incidence rates [8]. Moreover, the development of ML-based prognostic tool for CF can be a high-cost process that is not affordable in all countries where the disease is prevalent. This is because maintaining annual disease registries that collect data on individual patients requires an elaborate organizational effort and may entail a high cost per patient.

Instead of rebuilding ML-based prognostic models for each distinct patient population, models could be reused across populations. The extent to which it is appropriate to apply a model developed using data from one population in a demographically different population depends on whether the differences between the two populations undermine the model's prognostic accuracy. Models that exhibit high accuracy when tested on an external data set are likely to generalize well across populations, which could enable transferring models developed

in countries with well-maintained disease registries to other countries and populations where data are too scarce to develop a population-specific model.

In this paper, we investigate the external validity of ML-based prognostic models using annual follow-up data from the UK and Canadian Cystic Fibrosis Registries. In particular, we study the effect of (1) natural variations in patient characteristics across populations and (2) differences in clinical practice on the prognostic accuracy of ML-based prognostic scores. We use a state-of-the-art automated ML (AutoML) framework *AutoPrognosis* [9–12] to derive a prognostic score using data from the UK registry and validate the derived score using the Canadian registry data. We tested the applicability of the resulting ML-based LTx referral policy in different populations within the external validation data set and identified risk factors and patient subgroups associated with cross-population variations.

## Materials and methods

### Data and experiment setup

Annual follow-up data between 2008 and 2018 from the UK and Canadian Cystic Fibrosis Registries were used for analysis in the experiments. Both databases provided longitudinal records of CF patients, including variables on demographics, genetic mutation types, microbiology infection, medication and CF-related treatments, hospitalization, survival and transplantation status. The composite endpoint of LTx or death in a three-year horizon was considered as the target of prediction in order to identify appropriate candidates for LTx referral. A prognostic model was built by AutoPrognosis for this task, and the LTx referral was determined based on the model output and a cutoff threshold developed on training samples. The AutoPrognosis model and associated referral policy were derived on the larger UK CF population. The diagnostic accuracy of AutoPrognosis-constructed model was evaluated against two $FEV_1$-based baselines on the internal validation cohort from UK with ten-fold cross-validation. For the test of cross-population applicability, in each fold, the model was also evaluated on the external validation cohort from the Canadian CF dataset.

For the three-year outcome prediction task, the latest available records in 2014 covering over 99% of registered CF patients with annual reviews in the UK and Canada were used for experiments. The list of 53 commonly available variables considered in this study can be found in Table 1. The definition of mutation category [13] considered in this study was provided in S1 Appendix. The $FEV_1$% predicted values from the past three years before 2014 were included to provide additional information on lung function evolution of CF patients. Pediatric patients were excluded from this study due to low incidence rate of adverse endpoints (LTx or death) considered in this study [14, 15]. The complete sample selection criteria were illustrated in S1 Fig.

After removal of pediatric patients and samples with missing values, records of 4,610 and 2,008 patients from the UK and Canadian CF datasets were involved in this study, respectively. Ranges of considered feature variables in the two selected CF cohorts from UK and Canada were presented in Table 2. Additionally, hospitalized patients in the considered UK CF cohort in 2014 had a median hospital stay of 15 days with the interquartile range (IQR) of 8–32 days. Similar length of hospital stay was observed in the Canadian cohort with the median of 14 (IQR: 7–29) days. For patients received IV antibiotics treatment at home, the total days of treatment had a median of 22 (IQR 14–40) days in the UK while the median in Canada was 20 (IQR: 14–35) days. Further details of the two datasets can be found in the corresponding annual reports [14, 15] from the UK and Canadian CF Registries.

**Table 1. Common feature variables considered in the study.**

| | |
|---|---|
| Age | Gender |
| Height | Weight |
| BMI | FEV$_1$ (2014) |
| FEV$_1$% (2014) | FEV$_1$% (2013) |
| FEV$_1$% (2012) | FEV$_1$% (2011) |
| Aspergillus | Burkholderia Cepacia |
| Methicillin-resistant Staphylococcus aureus (MRSA) | Pseudomonas |
| Oxygen Therapy | Home IV Antibiotics Days |
| Hospitalization Days | Ivacaftor |
| HyperSaline | Inhaled Colistin |
| Chronic Macrolide | Cortico Oral |
| Cortico Inhaled | Cortico Combo |
| Antifungals | High-dose (HD) Ibuprofen |
| Allergic Bronchopulmonary Aspergillosis (ABPA) | Hemoptysis |
| Pneumothorax | Sinus Disease |
| Pancreatitis | Intestinal Obstruction |
| Cancer | Bone Fracture |
| Bone Loss | Depression/Anxiety |
| Liver Cirrhosis | Pancreatic insufficiency |
| Mutation Category {A, B, C, D, O} × {A, B, C, D, O} | |

## Population-level variations

Both UK and Canada are developed countries with publicly funded healthcare systems, which potentially implied good generalizability of ML models across their populations. However, while the nation-wide lung allocation scheme [16] prioritized organ offers to high-risk patients, in the UK, the final decision of LTx were mostly at the discretion of individual transplant center [17]. In Canada, without a national policy of lung resource allocation, offers of donor lungs were majorly determined by a *status* system based on the subjective clinical assessment by transplant centers [18]. Thereby, variations in LTx practice among the two populations are inevitable, which would impair the cross-population applicability of ML models. Further, we noted two major variations in patients' health status and LTx access that may impact the external validity of ML models developed from UK CF population when applied in Canada.

As reported in Table 2, the two studied cohorts had close variable ranges in demographics and lung functionality. The length of hospital stay and intravenous (IV) antibiotic treatment at home was similar for relevant patients in these two populations as well. However, the incidence rates of oxygenation, home IV antibiotic treatment, and hospitalization of Canadian CF patients were significantly lower compared to those in the UK, which could be a reflection of the approximate ten-year advantage in predicted survival of Canadian CF patients over the UK and other countries in 2014 [14, 15, 19, 20]. Such a gap in expected survival time was associated with shifts in the mortality rate distribution over populations as demonstrated in S2 Fig, which would impact the performance of AutoPrognosis-based policy on the external validation set.

Variation in LTx access was another important factor that impacted the applicability of AutoPrognosis-derived referral policy in the external Canadian CF cohort. Between 2008 and 2018, the UK had a median LTx per million population (PMP) of 2.90 with IQR between 1.60 and 3.08, while in Canada, the median LTx PMP was 6.49 (IQR: 5.44–8.10) [21]. The

**Table 2. Major characteristics of the studied UK and Canadian CF cohorts in 2014.** Binary variables were marked by * with occurrence and incidence rate reported. Continuous variables were reported with median value and IQR. Variables with different mean values in these two populations were identified via two-sample t-test under the p-value of 0.05 and were marked with †. Binary variables with a gap over 10% in incidence rate between the two populations were highlighted in bold.

| | Variable | UK | Canada | p-value |
|---|---|---|---|---|
| Demographics | Age | 29.67 (14.42) | 29.17 (15.18) | 0.8091 |
| | Male* | 2492 (54.06%) | 1086 (54.08%) | 0.9837 |
| | Female* | 2118 (45.94%) | 922 (45.92%) | 0.9837 |
| | Height† | 167.00 (14.00) | 167.00 (13.53) | 0.0437 |
| | Weight† | 61.00 (17.60) | 61.80 (17.31) | 0.0059 |
| | BMI† | 21.80 (4.44) | 22.18 (4.33) | 0.0000 |
| | FEV$_1$%† | 69.10 (37.06) | 66.24 (34.03) | 0.0001 |
| | Insufficiency Allele† | 1.96 (0.06) | 1.96 (0.07) | 0.0000 |
| Treatment | Oxygen Therapy*† | 353 (7.66%) | 68 (3.39%) | 0.0000 |
| | **IV Antibiotic Home*†** | 1703 (**36.94%**) | 344 (**17.13%**) | 0.0000 |
| | **Hospitalization*†** | 2127 (**46.14%**) | 420 (**20.92%**) | 0.0000 |
| | Ivacaftor*† | 255 (5.53%) | 51 (2.54%) | 0.0000 |
| | HyperSaline*† | 1452 (31.50%) | 509 (25.35%) | 0.0000 |
| | Inhaled Colistin*† | 2595 (56.29%) | 25 (1.25%) | 0.0000 |
| | **Chronic Macrolide*†** | 2588 (**56.14%**) | 792 (**39.44%**) | 0.0000 |
| | Cortico Oral*† | 501 (10.87%) | 123 (6.13%) | 0.0000 |
| | Cortico Inhaled*† | 581 (12.60%) | 112 (5.58%) | 0.0000 |
| | **Cortico Combo*†** | 2115 (**45.88%**) | 12 (**0.60%**) | 0.0000 |
| | Antifungals*† | 433 (9.39%) | 7 (0.35%) | 0.0000 |
| | HDI Buprofen*† | 1 (0.02%) | 5 (0.25%) | 0.0047 |
| Comorbidity | Liver Cirrhosis* | 328 (7.11%) | 120 (5.98%) | 0.0900 |
| | **ABPA*†** | 679 (**14.73%**) | 41 (**2.04%**) | 0.0000 |
| | **Hemoptysis*†** | 612 (**13.28%**) | 18 (**0.90%**) | 0.0000 |
| | Pneumothorax*† | 39 (0.85%) | 6 (0.30%) | 0.0128 |
| | **Sinus Disease*†** | 598 (**12.97%**) | 463 (**23.06%**) | 0.0000 |
| | Pancreatitis*† | 45 (0.98%) | 31 (1.54%) | 0.0463 |
| | Intestinal Obstruction* | 348 (7.55%) | 127 (6.32%) | 0.0761 |
| | Cancer*† | 18 (0.39%) | 23 (1.15%) | 0.0003 |
| | Fracture* | 35 (0.76%) | 16 (0.80%) | 0.8723 |
| | **Bone Loss*†** | 1221 (**26.49%**) | 122 (**6.08%**) | 0.0000 |
| | Depression/Anxiety*† | 298 (6.46%) | 226 (11.25%) | 0.0000 |
| Genetics | Mutation Category AB* | 826 (17.92%) | 351 (17.48%) | 0.6688 |
| | Mutation Category BB*† | 2416 (52.41%) | 945 (47.06%) | 0.0001 |
| | Mutation Category BC*† | 526 (11.41%) | 178 (8.86%) | 0.0020 |
| | Mutation Category BO*† | 354 (7.68%) | 210 (10.46%) | 0.0002 |
| Microbiology | Burkholderia Cepacia* | 223 (4.84%) | 86 (4.28%) | 0.3257 |
| | **Pseudomonas*†** | 3305 (**71.69%**) | 1103 (**54.93%**) | 0.0000 |
| | MRSA*† | 155 (3.36%) | 163 (8.12%) | 0.0000 |
| | Aspergillus*† | 979 (21.24%) | 551 (27.44%) | 0.0000 |

availability of lung resources was directly reflected in the LTx practice in these two countries. Among the studied UK CF patients in 2014, only 1.69% received LTx between 2015 and 2017, while 4.33% of the involved Canadian CF patients received LTx in the same period. Thereby, the cutoff threshold determined on the UK cohort could be over stringent when applied to the Canadian population.

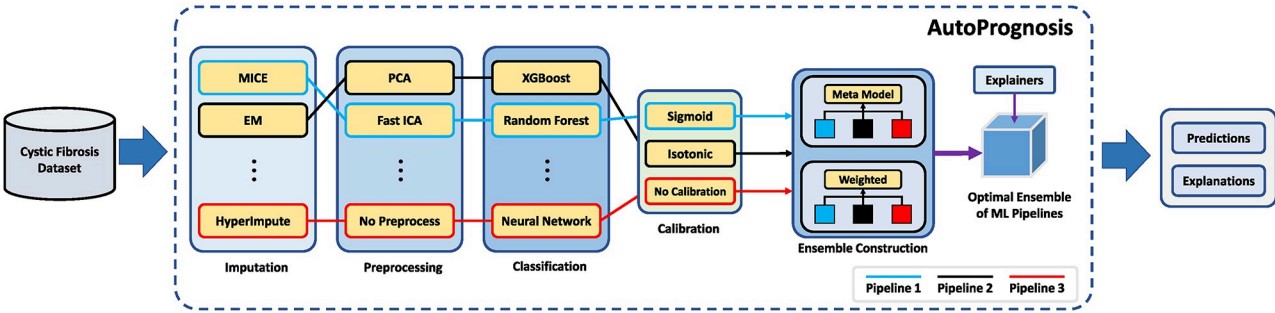

(a) AutoML workflow of the AutoPrognosis framework.

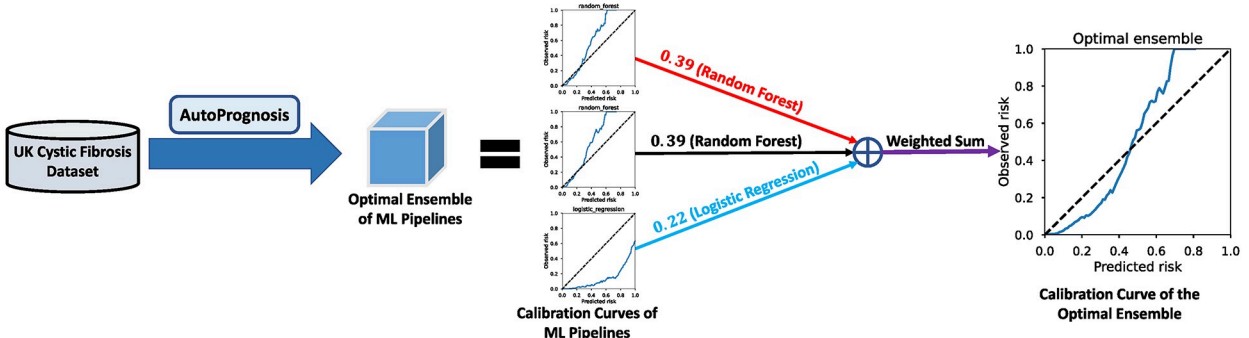

(b) Optimal ensemble constructed by AutoPrognosis on the UK CF cohort.

**Fig 1. An overview of the AutoPrognosis framework.** AutoPrognosis is a highly extensible AutoML framework built upon a plugin system. Based on the configured plugins for data imputation, preprocessing and classification, AutoPrognosis constructs an ML pipeline ensemble from the most performant pipelines developed with base classification plugins. (a) An example ensemble composed of three ML pipelines was illustrated to demonstrate the AutoML workflow of AutoPrognosis. All pipelines include four major procedures: imputation, preprocessing, classification, and calibration. In pipeline 1, the multivariate imputation by chained equations (MICE) plugin is applied for missing data imputation. The imputed data are then passed to fast ICA to create a compact, low-dimension data representation. The random forest classifier is used for the prediction task and its outputs are calibrated with a sigmoid function. Pipeline 2 and 3 are constructed in the same manner for the end-to-end prediction. AutoPrognosis first searches for the most performant ML pipelines among all possible combination of configured plugins. The selected pipelines are then combined as an ensemble model to achieve the best prediction performance. Two types of ensemble structure, i.e., stacked and weighted ensembles, are considered in AutoPrognosis, and Bayesian optimization is used to tune ensemble parameters for each structure. The optimal ensemble is selected based on the configured performance metric. Various explainer plugins of AutoPrognosis can be enabled for the ensemble to provide explanations along with the classification outputs. Detailed description of the algorithm can be found in [10–12]. (b) In this study, the UK CF dataset was provided as input to AutoPrognosis to search the optimal ML model for the composite endpoint prognostic task. The constructed ML model was a weighted ensemble of three ML pipelines. As illustrated in the calibration curves, the random forest pipelines tended to underestimate (above the dashed line) the risk level of high-risk patients. While the logistic regression pipeline was able to identify high-risk CF patients, its prognostic output was significantly higher than observed risks and would lead to many false alarms. AutoPrognosis was able to take advantage of all ML pipelines and create an optimal ensemble with the best prognostic accuracy.

## Methods

For the prognostic task considered in this paper, AutoPrognosis 2.0 [11, 12] is used to search for the optimal prognostic model as an ensemble of multiple ML pipelines as illustrated in the overview in Fig 1. AutoPrognosis 2.0 is an enhanced version of the original AutoPrognosis approach proposed in [10]. For the sake of convenience, we omit the version number and refer to AutoPrognosis 2.0 with AutoPrognosis in the following discussion. AutoPrognosis is to date the only open-source AutoML framework tailored for healthcare studies [12] and is validated to outperform existing AutoML methods [10, 11]. The end-to-end design and rich functionalities of the AutoPrognosis framework make it a convenient tool for clinical model

development and allow healthcare professionals to take advantage of state-of-the-art AutoML algorithms in their research without the requirement of extensive knowledge on machine learning [11].

Given an input dataset, AutoPrognosis automatically constructs ML pipelines as combinations of data imputation, preprocessing, and prediction modules in an end-to-end fashion. After calibration of the output function, these ML pipelines are combined as an ensemble for the best prognostic performance. Existing AutoML search algorithms including the default Bayesian optimization method can be utilized by AutoPrognosis to determine the optimal ensemble and ML pipeline configurations [11]. Further technical details can be found in the open-source software package [12]. In complementary to the development of optimal prognostic models, AutoPrognosis also provides clinical investigation functionalities to help clinicians further understand the derived prognostic model via explanation and cohort analysis modules. These inspection tools are specifically tailored for healthcare studies and enable clinicians to gain better understanding on feature variables and their impact on ML model predictions [11].

In this study, there was no missingness in both the UK and Canadian CF datasets after the sample selection and data preparation procedures described in S1 Fig and S1 Appendix. Thus, the data imputation plugin of AutoPrognosis was kept as the default imputation by chained equations (ICE) approach and was not actually applied to the input data. For the preprocessing step, normal, uniform, min-max, and identical transforms were configured for feature scaling, while the variance threshold, principal component analysis (PCA), and fast independent component analysis (ICA) plugins were used for dimensionality reduction. Base models of neural network, XGBoost, random forest, and logistic regression were configured as classification plugins for the pipeline construction. The optimal ensemble was selected via ten steps of Bayesian optimization.

## Statistical analysis

For the evaluation of prognostic accuracy, precision, recall, F1, area under the curve of receiver-operating characteristic (AUCROC), and area under the curve of precision-recall (AUCPRC) about the composite endpoint of LTx or death were used as performance metrics. Given a diagnostic model for adverse endpoint estimation, the precision score, i.e., positive predictive value (PPV), is calculated as the proportion of true high-risk patients experienced adverse outcomes among all patients identified as high-risk by the model. The recall score, i.e., true positive rate (TPR), is the fraction of true high-risk patients that were correctly identified by the model. The F1 score is the harmonic mean of precision and recall scores, which reflects the overall diagnostic power of a model. In this paper, all of these performance metrics were evaluated via endpoint-stratified ten-fold cross-validation on the source dataset of UK CF cohort for the 95% confidence interval (CI). The Canadian CF cohort was used for external validation in each fold.

## Ethics statement

This study was conducted within the scope of 'Developing a prognostic score for people with cystic fibrosis using machine learning' which was approved by the Cambridge Psychology Research Ethics Committee of the University of Cambridge (Application No. PRE.2021.034). Our access to registry data of CF patients from the UK Cystic Fibrosis Registry and Cystic Fibrosis Canada was approved by the Registry Research Committee or Registry Review Panel comprised of CF clinicians and researchers. The registry data collected by the UK Cystic Fibrosis Registry included demographic, clinical treatment and outcome information from patients

that provided written consents. The data from Cystic Fibrosis Canada were collected with consent from individual patient through accredited CF clinics in Canada.

## Results

### Prognostic power of LTx referral policies

The prognostic performance of the model constructed by AutoPrognosis was evaluated against the baseline LTx referral policy of $FEV_1 \leq 30\%$ predicted introduced in [6] and the policy derived from a recent guideline [22] proposed by the Cystic Fibrosis Foundation. This guideline recommended LTx referral for adult patients in the following cases: 1) $FEV_1$ below 50% predicted with rapid decline in the past 12 months; 2) $FEV_1$ below 40% predicted with markers of reduced survival time [22–24]; or 3) $FEV_1$ below 30% predicted. Detailed criteria in the adapted referral policy were provided in S1 Appendix. The AutoPrognosis policy was developed from the prognostic model constructed by AutoPrognosis by applying a cutoff threshold to the model output. Patients with predicted risk score above the threshold were recommended for LTx referral. The cutoff threshold was chosen to maximize the F1 score on training samples of AutoPrognosis. In the meantime, comparison of the two populations from the UK and Canada showed variations in LTx rates in certain patient subgroups as reported in S1 Table. Two additional selection criteria ($FEV_1 \leq 30\%$ predicted or $FEV_1 \leq 40\%$ predicted with an absolute decrease of $\Delta FEV_1 \geq 10\%$ over the past three years) were introduced based on this observation to construct an augmented AutoPrognosis policy.

As shown in Table 3, the ML-based diagnostic model constructed via AutoPrognosis had much better accuracy over the baseline referral policy of $FEV_1 \leq 30\%$ on UK cohort, which was consistent with the result reported in [6]. The referral policy derived from the 2019 guideline [22] was capable to select most of the high-risk patients and had a higher TPR of 0.63 compared to AutoPrognosis. However, there were many false alarms in its high-risk patient selection given the lowest PPV score of 0.31 on the internal validation set. Due to the limited lung resources in the UK [21], high PPV score is essential for any practical LTx referral policies. The AutoPrognosis policy was validated to be optimal on the studied UK CF cohort with the highest F1 score of 0.49. In addition to the population-level performance metrics, the

**Table 3. Diagnostic performance of four LTx referral policies.** The policy of $FEV_1 \leq 30\%$ predicted and a $FEV_1$-based policy derived from a 2019 guideline were used as two baselines. Apart from the original AutoPrognosis policy developed on the studied UK CF cohort, an augmented AutoPrognosis policy with two additional criteria developed from the Canadian CF data was included in the evaluation as well. As a balanced measurement of prognostic accuracy, a high F1 score comes with high PPV and TPR scores. Decrease in either PPV or TPR score leads to a lower F1 score. Desirable LTx referral policies ought to have a high F1 score. All evaluation results were reported with 95% CI.

| Prognostic performance | | $FEV_1 \leq 30\%$ | Guideline 2019 [22] | AutoPrognosis | |
|---|---|---|---|---|---|
| | | | | Original | Augmented |
| Internal validation (UK cohort) | PPV | 0.46±0.04 | 0.31±0.02 | 0.50±0.05 | 0.37±0.02 |
| | TPR | 0.39±0.05 | 0.63±0.05 | 0.49±0.05 | 0.61±0.05 |
| | F1 | 0.42±0.04 | 0.41±0.02 | 0.49±0.04 | 0.46±0.02 |
| | AUCROC | – | – | 0.91±0.01 | – |
| | AUCPRC | – | – | 0.49±0.05 | – |
| External validation (Canadian cohort) | PPV | 0.47±0.00 | 0.35±0.00 | 0.52±0.02 | 0.42±0.00 |
| | TPR | 0.31±0.00 | 0.53±0.00 | 0.24±0.02 | 0.49±0.00 |
| | F1 | 0.37±0.00 | 0.43±0.00 | 0.33±0.02 | 0.45±0.00 |
| | AUCROC | – | – | 0.88±0.00 | – |
| | AUCPRC | – | – | 0.40±0.00 | – |

calibration curves of the ML model constructed by AutoPrognosis on the UK and Canadian CF cohorts were provided in S3 Fig to illustrate its accuracy for patients with different risk levels.

The cross-population applicability of these LTx referral policies was validated on the external Canadian CF cohort. Benefited from the widely accepted referral criteria, the guideline-derived policy had similar PPV and TPR scores in both internal and external validations, which was within expectation. Meanwhile, despite the high PPV score of 0.52, majority (76% according to the TPR of 0.24) of high-risk patients in Canadian CF population was overlooked by the original AutoPrognosis policy, which led to the lowest TPR score among all evaluated policies. Similar failure in external validity of AutoPrognosis-based policy was observed even when only the single endpoint of death (without LTx) was considered as presented in S1 Appendix. Such failure can be overcame via accounting for the variations in LTx practice existed in certain subgroups of CF patients. According to Table 3, with merely two additional selection criteria, the augmented AutoPrognosis policy achieved better prognostic accuracy (F1) than the 2019 guideline on the external validation set from Canadian CF population. The TPR score of the original AutoPrognosis policy was boosted from 0.24 to 0.49 without much loss in PPV. In comparison, although the 2019 guideline had a high TPR score of 0.53, the lower PPV of 0.35 showed that there were 65% of false alarms in its LTx referral.

## Timeliness of LTx referral policies

As a complement to the diagnostic accuracy (TPR) reported in Table 3, the timely LTx referral of high-risk CF patients was further evaluated as shown in Table 4. The LTx referral for a high-risk patient with adverse future endpoint in a three-year horizon was considered in time. Timely LTx referral of high-risk CF patients is especially important when there are sufficient donor lung resources available. Failure in identifying these patients leads to delayed referrals which may leave significant impact on their life expectancy. While the AutoPrognosis-based policy was as effective as the 2019 guideline in issuing timely LTx recommendation on the UK cohort, it resulted in a significantly larger number of delayed LTx referrals compared to the $FEV_1$-based baselines when applied to the Canadian CF population, which is unacceptable in practical applications. In contrast, with two additional referral criteria for two subgroups of moderate- and high-risk patients, the augmented AutoPrognosis policy was as timely as the 2019 guideline while having a significantly higher precision (PPV) score for both cohorts as shown in Table 3.

## Risk factors across populations

To better understand the decreased accuracy of AutoPrognosis-constructed model on the external validation set from Canada, the prognostic power of individual variable was compared

**Table 4. Timeliness of referral policies in target domain.** There were 307 and 158 patients died or received LTx during 2015 and 2017 in the considered UK and Canadian CF cohorts, respectively. A timely referral policy shall suggest LTx referral for these patients based on the annual follow-up information in 2014. The referral result of the baselines and AutoPrognosis model developed on the source data (UK cohort) were evaluated on such subgroup of CF patients. For each policy applied to this subgroup, patients predicted as low-risk were counted under the column of *delayed*. Patients predicted to be high-risk were counted in the column of *timely*. The numbers were reported with 95% CI via models obtained from stratified ten-fold cross-validation on the UK CF dataset.

| Referral policy | | UK cohort | | Canadian cohort | |
|---|---|---|---|---|---|
| | | **Delayed** | **Timely** | **Delayed** | **Timely** |
| $FEV_1 \leq 30\%$ | | 187.00±0.00 | 120.00±0.00 | 109.00±0.00 | 49.00±0.00 |
| Guideline 2019 | | 114.00±0.00 | 193.00±0.00 | 74.00±0.00 | 84.00±0.00 |
| AutoPrognosis | Original | 134.10±6.77 | 172.90±6.77 | 120.10±3.39 | 37.90±3.39 |
| | Augmented | 103.70±4.66 | 203.30±4.66 | 80.30±0.62 | 77.70±0.62 |

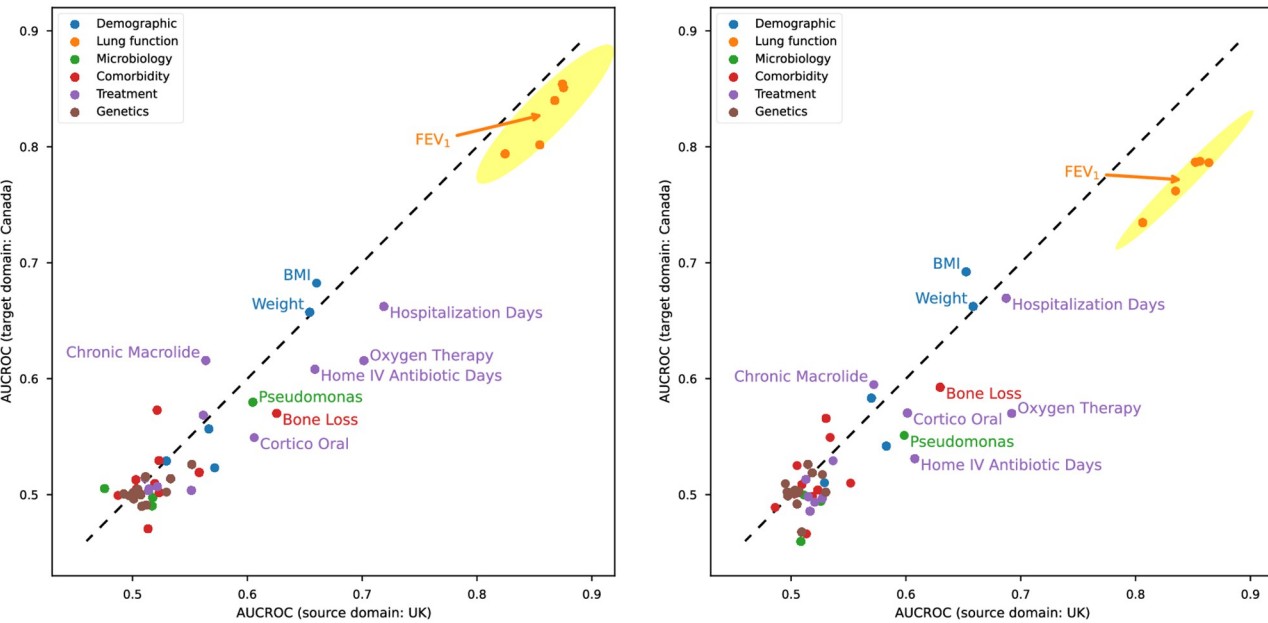

(a) Composite endpoint of LTx or death    (b) Single endpoint of death without LTx

**Fig 2. Diagnostic accuracy of individual variables in the UK and Canadian CF cohorts.** The ML model constructed by AutoPrognosis was trained with one single feature variable as input iteratively on the UK cohort. The AUCROC score was used as the proxy of diagnostic accuracy and was measured with ten-fold cross-validation. The Canadian CF cohort was used as the external validation set in each fold. Feature variables were colored based on their category, and their locations were determined by the average AUCROC score achieved by their associated models on the two populations. Feature variables with AUCROC score above 0.6 [25] were considered to be predictive of high-risk patients and were annotated with their variable names.

in the two studied populations. The comparison was conducted for both the composite endpoint and the single endpoint of death to differentiate the impact of LTx access and shift in mortality rate across populations. The result was illustrated in Fig 2. Variables close to or above the dashed line were shared risk factors across populations since they achieved equal or higher accuracy as in the UK population when applied to the Canadian CF cohort.

According to [23], $FEV_1$ score (current and in the past three years), body mass index (BMI), hospitalization, IV antibiotics treatment, oxygen therapy, oral corticosteroids are strong predictors for death or LTx in a three-year horizon. Further, underweight (low BMI) is an important factor that affects the listing and ranking of patients waiting for LTx [26–29]. The variable of weight plays a similar role as BMI in the prognostic task since the height of adult patients were generally stable. Pseudomonus infection is correlated to CF patients' mortality [30, 31], and long-term macrolide therapy is in widespread use for CF patients, particularly for the treatment of chronic pseudomonus infection [32]. In the meantime, long-term oral corticosteroids are usually used for CF patients with advanced lung diseases or allergic bronchopulmonary aspergillosis (ABPA) [33]. While ABPA only causes death to CF patients in some rare cases [34], advanced lung diseases and the associated risk of respiratory failure are closely related to the adverse endpoints considered in this study. The bone mineral loss, i.e., osteopenia (mild) or osteoporosis (severe), is suffered by certain subgroups of CF patients and is associated with the severity of inflammatory lung damage, which would ultimately lead to death or LTx referral, of these patients [35, 36].

As shown in Fig 2(a), most of the risk factors discussed above were shared by the UK and Canadian CF populations, which suggested good generalizability of the AutoPrognosis model

developed on UK CF population when applied to the Canadian CF cohort. However, variables located under the dashed line, i.e., hospital stay, oxygenation, treatment of IV antibiotics at home, medication of corticosteroids and, the indicator of bone loss, were less predictive of LTx or death in the Canadian CF cohort. The reason behind this was twofold. On one hand, along with the better survival expectation [14, 15], patients in Canada had much lower incidence rates of bone loss, hospitalization, IV antibiotics treatment and oxygenation as reported in Table 2 and S6 Fig. Such distributional shifts in population capped the upper bounds in predictive accuracy of these variables. On the other hand, the higher donor lung availability [21, 37] potentially allowed less severe CF patients in Canada to receive LTx, which resulted in the lower relevance between LTx and these markers of severe lung diseases. For instance, the risk factor of hospital stay got closer to the dashed line when only the single endpoint of death was considered in Fig 2(b), which indicated nearly equal predictive power in the two populations compared to Fig 2(a).

As a direct measure of lung function, $FEV_1$ is the most important risk factor for both the composite endpoint and the single endpoint of death. However, Fig 2(b) showed that it had lower diagnostic accuracy on death when evaluated on the external validation set from Canadian CF population. This was linked to the shifts in mortality distribution across the two populations. As presented in S2 Fig, for Canadian CF patients with $FEV_1$ below 30% predicted, the observed mortality risk was significantly lower than those in the UK. While in Fig 2(a) with LTx considered in the endpoint, the drop in predictive accuracy of $FEV_1$ was compensated due to higher LTx rate for this subgroup of patients in Canada according to S1 Table.

## Model applicability in subgroups

We further examined the prognostic errors of the AutoPrognosis-constructed model in several subgroups of Canadian CF patients. For this evaluation, three risk groups (low-, moderate-, and high-risk) were defined based on the incidence rate intervals in $\mathcal{R} = \{[0, 0.1), [0.1, 0.5), [0.5, 1.0]\}$ for adverse outcomes in the future. High-risk patients were considered to have higher chances of death or LTx while low-risk patients were unlikely to receive LTx referral. The risk score from AutoPrognosis output was translated into risk group labels based on the risk group cutoffs, which were searched via the observed risk among patients with similar predicted risk scores. For illustrative purpose, the stratification result in the UK CF cohort was plotted in S4 Fig.

The subgroup-level applicability of the AutoPrognosis model developed on the UK CF population was evaluated via its prognostic consistency with a separate model constructed by AutoPrognosis on the Canadian cohort, as shown in Fig 3. The risk group division developed from UK cohort was applicable to 86.3% of studied CF patients from Canada, especially for the patients with very high or low risks.

In area 1 of mismatch, adverse future outcomes were observed for 22.9% patients, indicating moderate-risk as the correct label. Further, there was a subset of patients with $FEV_1$ below 40% predicted and an absolute decline of $\Delta FEV_1 \geq 10\%$ predicted over the past three years, which showed a clear sign of lung function deterioration. Patients with such patterns in Canada belonged to the moderate-risk stratum in general as illustrated in S5 Fig. We noted that over 95% of patients in this area had $FEV_1$ above 30% predicted. Since $FEV_1$ below 30% predicted was reported to be a major risk factor for LTx referral in the UK [6], the low-risk stratum wrongly assigned to this area could be potentially explained with the higher $FEV_1$ level of patients in this area.

Regarding the area 2 of mismatch, although only 30.7% of associated patients were oxygenated, this area had an observed risk of adverse future outcome at 68.0% and was qualified for

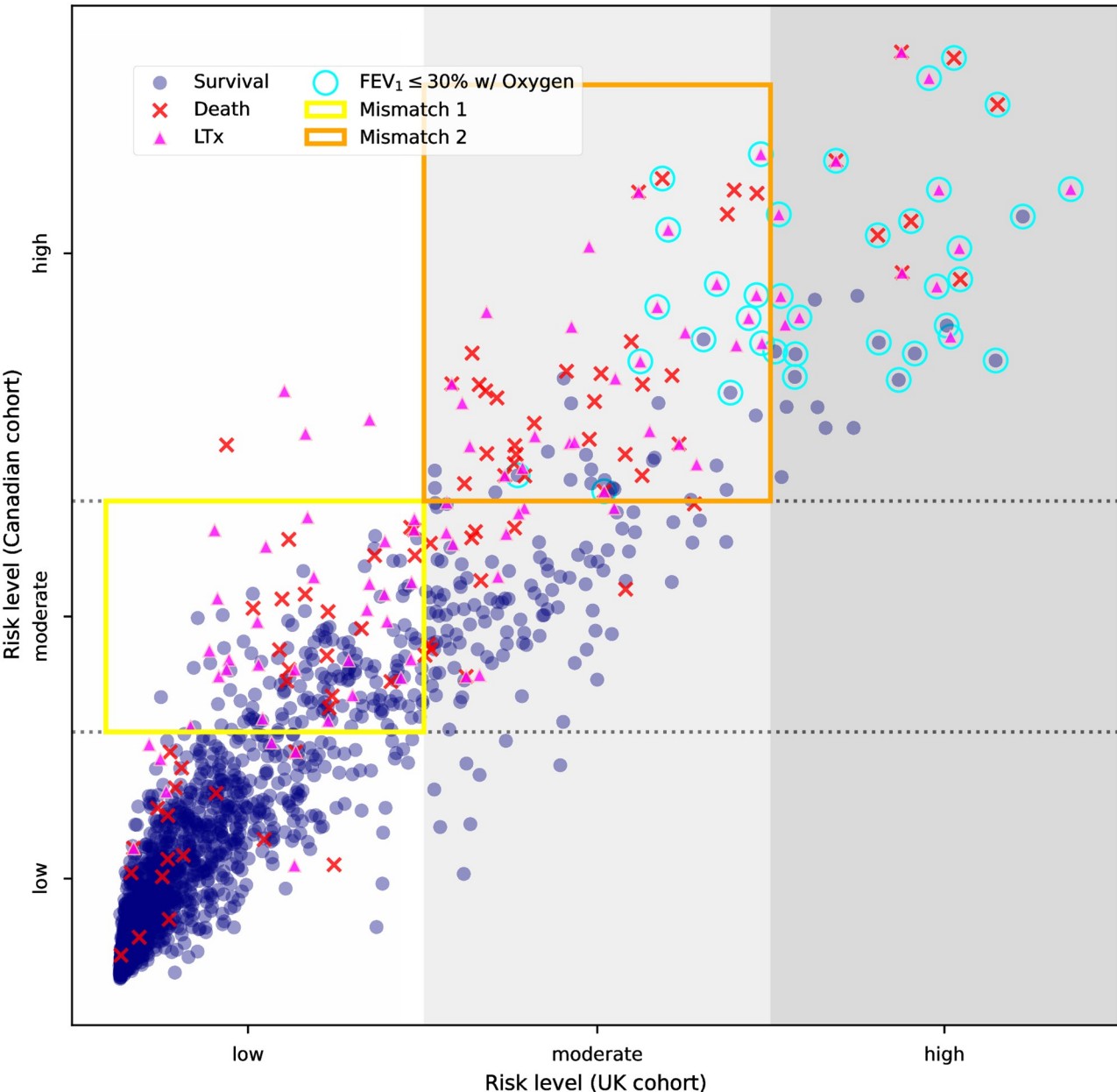

**Fig 3. Mismatches in risk stratification between the UK and Canadian CF cohorts.** Two prognostic models were constructed by AutoPrognosis separately on the UK and Canadian CF populations. Canadian CF patients with future outcomes of survival, death and LTx were annotated with circles, crosses, and triangles, respectively. Their locations were determined by the output of the two AutoPrognosis models. Very high-risk patients with $FEV_1$ below 30% predicted and supplement oxygenation were highlighted with cyan circles. The two AutoPrognosis models were in agreement of risk stratum for most of these very high-risk patients. Mismatches happened in two subgroups of patients with underestimated risk levels. The first subgroup consisted of moderate-risk patients that were identified as low-risk by the AutoPrognosis model developed on the UK population and was referred to as mismatch 1. The latter subgroup consisted of high-risk patients that fell into the moderate-risk stratum in the UK population. We referred to the corresponding area as mismatch 2.

the high-risk stratum. Canadian CF patients in this area had a median $FEV_1$ of 29.8% predicted. Risk stratum mismatches in this area were majorly caused by 48.0% of the patients with $FEV_1$ above 30% predicted and 34.7% of patients with $FEV_1$ below 30% but received no supplement oxygen.

**Augmented policy for the external validation set**

Despite low TPR score of the original AutoPrognosis policy on the external validation set from Canada, the comparison of key risk factors and risk strata of these two populations showed that many risk factors were shared by the UK and Canadian CF populations and the AutoPrognosis model developed on UK CF cohort was applicable to most CF patients in Canada except for a few subgroups affected by the cross-region variations. In the following discussion, we showed that the adaptation with two additional criteria can significantly improve the diagnostic accuracy of the AutoPrognosis-based policy on the external validation set from Canada.

As discussed above, in the Canadian CF cohort, patients with $FEV_1$ below 30% predicted or $FEV_1$ below 40% predicted plus an absolute decline of $\Delta FEV_1 \geq 10\%$ in the past three years had a higher risk stratification compared to those in the UK and should be recommended for LTx referral when there were sufficient lung resources available. This can be verified with the distribution of patients in these two subgroups as provided in S5 Fig. Additionally, statistics in S1 Table showed that Canadian CF patients in these two specific subgroups had significantly higher rates of LTx compared to the UK cohort. Although S2 Fig showed that there existed a distribution shift in mortality rate for patients with $FEV_1$ below 30% predicted in these two populations, such shift was mostly compensated by the higher LTx rate in Canada, and no relevant overestimation in risk stratum was observed in Fig 3.

To account for the above-mentioned variation across populations, we proposed an augmented version of the AutoPrognosis policy by incorporating prediction of LTx referral for these two subgroups as additional criteria. According to the prognostic accuracy reported in Table 3, the augmented AutoPrognosis policy achieved an improved prognostic performance with PPV of 0.42 and TPR of 0.49 on the external validation set from Canadian CF population. The augmented policy outperformed the consensus LTx referral guideline [22] with a comparable TPR score and a significantly higher PPV score, which led to the best diagnostic accuracy (F1 of 0.45) over the original AutoPrognosis policy and two $FEV_1$-based baselines.

## Discussion

The clinical practice of LTx for CF patients is hugely affected by guidelines. Thereby, LTx prediction can be heavily biased in favor of current guidelines that focus on $FEV_1$. As illustrated in Fig 2, predictive power of individual risk factor may differ when different targets (LTx or death v.s. death without LTx) were considered. In this paper, we focused on the evaluation of external validity of ML models in a different population and analyzed the effect of various factors on the cross-population generalizability of ML models for LTx referral. The impact of guideline-induced biases is out of the scope of our paper, and we leave it as a future direction of our study.

Our study is based on the data obtained from the UK and Canadian Cystic Fibrosis Registries. While we have worked hard to remove possible biases from data processing procedures, the results and conclusion in this paper may be affected by potential errors in records in these two datasets. In the meantime, mismatches in variable definition across healthcare systems are usually inevitable. Their impact on prognostic biases of an ML model applied across populations could be entangled with other major factors like patient health status and organ availability analyzed in this manuscript. Risk factor analysis in this study shows that variables affected by such type of mismatches have little impact on the external validity of ML models in risk prognostication for CF patients. The impact attribution of different sources of prognostic biases across populations is an important topic, and we consider it as another possible direction of our future work.

## Conclusion

To validate the cross-population applicability of ML-based prognostic models for poor clinical outcomes of CF patients, we conducted a study on external validity of ML models and their derived LTx referral policies based on annual follow-up data from the UK and Canadian Cystic Fibrosis Registries. The impact of LTx access and distribution shifts in patients' health status on underlying risk factors and risk stratification was evaluated via the state-of-the-art AutoML framework AutoPrognosis. $FEV_1$ was verified to be the most significant risk factor for adverse outcome diagnostic of CF patients. Two $FEV_1$-defined subgroups of patients were identified to be hugely affected by the cross-population variations in the external validation set from Canada. Further analysis showed that appropriate consideration of these variation-associated subgroups helped to the adaptation of ML models in a different population. Our experiments highlighted the importance of external validation of ML models for CF outcome diagnostic. The uncovered insights on external validity can be used to guide the real-world adaptation of the high-precision ML models on different populations, and inspires new research on applying modern transfer learning methods for fine-tuning models in environments with significant variation in care and patient demographics.

## Supporting information

**S1 Fig. Flow charts of patients inclusion criteria.**
(TIF)

**S2 Fig. Comparison of $FEV_1$ and mortality rate distribution in the studied UK and Canadian CF populations.**
(TIF)

**S3 Fig. Calibration of the prognostic model developed on the considered UK CF cohort.**
(TIF)

**S4 Fig. Risk stratification developed on the UK CF cohort.**
(TIF)

**S5 Fig. Two subgroups of moderate- and high-risk CF patients from Canada.**
(TIF)

**S6 Fig. Comparison of important continuous variables in the considered adult UK and Canadian CF cohorts via box plot.**
(TIF)

**S1 Table. LTx rates in subgroups in the studied adult UK and Canadian CF cohorts.**
(PDF)

**S1 Appendix. Additional supplementary materials.**
(PDF)

## Acknowledgments

We acknowledge the UK Cystic Fibrosis Registry and Canadian Cystic Fibrosis Registry for the high quality data of CF patients. We thank Dr. Janet Allen (University of Cambridge), Ms. Stephanie Chang (Cystic Fibrosis Canada) and Prof. Anne Stephenson (University of Toronto) for their help with the CF data access, extraction and data correction. We would like to express special thanks to Prof. Changhee Lee (Chung-Ang University) for his help with the cleaning and preprocessing of the UK CF data.

## Author Contributions

**Conceptualization:** Yuchao Qin.

**Formal analysis:** Yuchao Qin, Ahmed Alaa, Andres Floto.

**Methodology:** Yuchao Qin, Ahmed Alaa.

**Supervision:** Mihaela van der Schaar.

**Visualization:** Yuchao Qin, Mihaela van der Schaar.

**Writing – original draft:** Yuchao Qin.

**Writing – review & editing:** Yuchao Qin, Ahmed Alaa, Andres Floto, Mihaela van der Schaar.

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
