## [Decision Letter · Decision Letter 0]

14 Nov 2022

PDIG-D-22-00234

External Validity of Machine Learning-based Prognostic Scores for Cystic Fibrosis: A Retrospective Study using the UK and Canadian Registries

PLOS Digital Health

Dear Dr. Qin,

Thank you for submitting your manuscript to PLOS Digital Health. After careful consideration, we feel that it has merit but does not fully meet PLOS Digital Health's publication criteria as it currently stands. Therefore, we invite you to submit a revised version of the manuscript that addresses the points raised during the review process.

Please submit your revised manuscript within 30 days Dec 14 2022 11:59PM. If you will need more time than this to complete your revisions, please reply to this message or contact the journal office at digitalhealth@plos.org. Please include the following items when submitting your revised manuscript:

We look forward to receiving your revised manuscript.

Kind regards,

Mecit Can Emre Simsekler, Ph.D.

Academic Editor

PLOS Digital Health

Journal Requirements:

1. Please ensure that you provide a single, cohesive .tex source file for your LaTeX revision. You may upload this file as the item type 'LaTeX Source File.' As stated in the PLOS template, your references should be included in your .tex file (not submitted separately as .bib or .bbl). Please also ensure that you are making any formatting changes to both your .tex file and the PDF of your manuscript. If you have any questions, please contact Latex@plos.org. You can find our LaTeX guidelines here: https://journals.plos.org/digitalhealth/s/latex

Additional Editor Comments (if provided):

Reviewers' comments:

Reviewer's Responses to Questions

**Comments to the Author**

1. Does this manuscript meet PLOS Digital Health’s publication criteria? Is the manuscript technically sound, and do the data support the conclusions? The manuscript must describe methodologically and ethically rigorous research with conclusions that are appropriately drawn based on the data presented.

Reviewer #1: Yes

Reviewer #2: Yes

2. Has the statistical analysis been performed appropriately and rigorously?

Reviewer #1: Yes

Reviewer #2: Yes

3. Have the authors made all data underlying the findings in their manuscript fully available (please refer to the Data Availability Statement at the start of the manuscript PDF file)?

Reviewer #1: Yes

Reviewer #2: Yes

4. Is the manuscript presented in an intelligible fashion and written in standard English?

Reviewer #1: Yes

Reviewer #2: Yes

5. Review Comments to the Author

Reviewer #1: Overall, this is a clear, concise, and well-written manuscript. The introduction is relevant and theory-based. Sufficient information about the previous study findings is presented for readers to follow the present study rationale and procedures. The methods are generally appropriate, although clarification of a few details and a justification for using this particular method of AutoML should be provided. Overall, the results are clear, and the comparison for model validation is good, however, more Datasets description is needed, also the balance and the potential bias from the mismatches between the variables of the datasets should be addressed and spotted as mentioned in the discussion section. Overall results were inciteful and represented a guide to the cross-population adaptation of ML-based models and the conclusion of applying transfer learning methods for fine-tuning ML.

Reviewer #2: I should say I was impressed with the quality of the manuscript. Authors have presented a rigorous study supported by extensive statistical analysis and impressive plots. I have no comments and I would recommend it for publication without any hesitation.

6. PLOS authors have the option to publish the peer review history of their article (what does this mean?). If published, this will include your full peer review and any attached files.

**Do you want your identity to be public for this peer review?** For information about this choice, including consent withdrawal, please see our Privacy Policy.

Reviewer #1: No

Reviewer #2: No

---

## [Editor Report · Decision Letter 1]

8 Dec 2022

External Validity of Machine Learning-based Prognostic Scores for Cystic Fibrosis: A Retrospective Study using the UK and Canadian Registries

PDIG-D-22-00234R1

Dear Mr Qin,

We are pleased to inform you that your manuscript 'External Validity of Machine Learning-based Prognostic Scores for Cystic Fibrosis: A Retrospective Study using the UK and Canadian Registries' has been provisionally accepted for publication in PLOS Digital Health.

Best regards,

Mecit Can Emre Simsekler, Ph.D.

Academic Editor

PLOS Digital Health